# Effects of Cerebrospinal Fluids from Alzheimer and Non-Alzheimer Patients on Neurons–Astrocytes–Microglia Co-Culture

**DOI:** 10.3390/ijms25052510

**Published:** 2024-02-21

**Authors:** Matilda Iemmolo, Giulia Bivona, Tommaso Piccoli, Aldo Nicosia, Gabriella Schiera, Carlo Maria Di Liegro, Fabrizio Di Pietra, Giulio Ghersi

**Affiliations:** 1Department of Biological, Chemical and Pharmaceutical Sciences and Technologies, 90123 Palermo, Italy; matilda.iemmolo@unipa.it (M.I.); gabriella.schiera@unipa.it (G.S.); carlomaria.diliegro@unipa.it (C.M.D.L.); fabriziodipietra@unipa.it (F.D.P.); 2Department of Biomedicine Neurosciences and Advanced Diagnostics, University of Palermo, 90127 Palermo, Italy; giulia.bivona@unipa.it; 3Department of Laboratory Medicine, University Hospital “P. Giaccone”, 90127 Palermo, Italy; tommaso.piccoli@unipa.it; 4Institute for Biomedical Research and Innovation—National Research Council (IRIB-CNR), 90146 Palermo, Italy; aldo.nicosia@cnr.it

**Keywords:** Alzheimer, CX3CL1, neuroinflammation, tau, β-amyloid

## Abstract

Alzheimer’s disease (AD) is the most common form of dementia, characterized by the accumulation of β-amyloid plaques, tau tangles, neuroinflammation, and synaptic/neuronal loss, the latter being the strongest correlating factor with memory and cognitive impairment. Through an in vitro study on a neurons–astrocytes–microglia (NAM) co-culture system, we analyzed the effects of cerebrospinal fluid (CSF) samples from AD and non-AD patients (other neurodegenerative pathologies). Treatment with CSF from AD patients showed a loss of neurofilaments and spheroids, suggesting the presence of elements including CX3CL1 (soluble form), destabilizing the neurofilaments, cellular adhesion processes, and intercellular contacts. The NAM co-cultures were analyzed in immunofluorescence assays for several markers related to AD, such as through zymography, where the expression of proteolytic enzymes was quantified both in cell extracts and the co-cultures’ conditioned medium (CM). Through qRT-PCR assays, several genes involved in the formation of β-amyloid plaque, in phosphorylation of tau, and in inflammation pathways and MMP expression were investigated.

## 1. Introduction

Alzheimer’s disease (AD) is the most common form of dementia, affecting nearly 45 million people worldwide. The main neuropathological features of AD are the accumulations of β-amyloid plaques, tau tangles, neuroinflammation, synaptic dysfunction, and neuronal loss, the latter being the strongest correlating factor with memory and cognitive impairment. Many of these pathological signs influence each other during the onset and progression of the disease. Recent genetic evidence suggests the possibility of a causal link between impaired immune pathways and synaptic dysfunction in AD. Emerging studies also suggest that immune-mediated synaptic pruning could initiate the early-stage pathogenesis of AD [1].

CX3CL1 is a chemotactic cytokine that influences Alzheimer’s pathology. It is composed of 373 amino acids and is functionally divided into four domains: an extracellular domain of 76 amino acids linked to an extended stem of 241 amino acids like mucin, followed by transmembrane and intracellular domains of 21 and 35 amino acids, respectively. CX3CL1 is a chemotactic cytokine widely expressed by neurons within the hippocampus and cortex. It is a transmembrane protein that exists both in the membrane-associated form of neurons and astrocytes and in soluble form (sCX3CL1). The former acts as an adhesion molecule and, after enzymatic processing, a signaling one, interacting with its unique CX3CR1 receptor, present on the microglia. The soluble form is generated through a proteolytic process, which plays a significant role in acute inflammation [2].

CX3CL1/CX3CR1 signaling mediates neuron–glia communication by keeping microglia in a resting state (quiescent or inactivated), thereby inhibiting the release of proinflammatory cytokines. Furthermore, it enhances the phagocytosis of pre-synaptic elements [3]. CX3CL1/CX3CR1 signaling can inhibit inflammation and tau phosphorylation in AD, but it still results in an increase in the deposition of β-amyloid fragments. It has been suggested that at the onset of AD, the intra-neuronal accumulation of β-amyloid causes a slight decrease in CX3CL1/CX3CR1 signaling, resulting in increased Aβ phagocytosis and the hyperphosphorylation of tau [4]. However, CX3CL1-mediated neuron–glia crosstalk in the context of AD has reported conflicting results [5,6]. The CX3CL1/CX3CR1 interaction plays an important role in maintaining a healthy and anti-inflammatory condition in the brain. CX3CL1’s ability to be both an adhesion molecule and a soluble element makes it likely to acquire a signaling function. However, it is not yet clear, for soluble CX3CL1, how and how many forms are generated enzymatically during the dispersion of the molecule and what changes are induced in the metabolic pathways after binding to its CX3CR1 receptor. Some transcription factors (P38, β-catenin, NF-kB) and associated molecules (AKT and GSK3-β) have been linked to AD. The activation of p38 has been reported to occur in the early stages of AD. β-amyloid plaques have been found to stimulate microglia towards the rapid activation of MAPK p38, resulting in the upregulation of proinflammatory cytokines such as IL-1 and TNFα. Furthermore, IL-1 activates MAPK p38 in astrocytes and neurons, causing excessive inflammation and the phosphorylation of tau [7]. The peptides have also been reported to stimulate microglia to express high levels of IL-1, IL-6, and IL-18. The production of proinflammatory cytokines induces the activation of astrocytes followed by microgliosis [8]. Another AD-related pathway is the activation of Wnt/β-catenin signaling, which inhibits the production of amyloid peptides and the hyperphosphorylation of brain tau protein. An increase in the Wnt antagonist DKK1 causes a rapid decrease in the levels of essential signaling proteins including β-catenin, cyclin D1, c-myc, Wnt7a, and PSD95 [9]. Another important hallmark of AD is the presence of intracellular tangles composed of hyperphosphorylated forms of the microtubule-associated protein tau in neurons. GSK3 is an important kinase associated with the hyperphosphorylation of the tau protein at the phosphorylation sites relevant to AD. The activation of Wnt/β-catenin signaling results in the inhibition of GSK3 activity and subsequent suppression of tau phosphorylation. Furthermore, the activation of Wnt/β-catenin signaling can inhibit tau hyperphosphorylation and neuronal death [9]. Apolipoprotein E4 (ApoE4) can inhibit Wnt/β-catenin signaling [9]. Furthermore, NF-kB is involved in neurodegenerative diseases since it activates ROS, characteristic elements of patients with AD [10,11].

As previously mentioned, CX3CL1, in addition to being an adhesion molecule, exists in the soluble form, sCX3CL1. The soluble form is generated through the action of metalloproteases (MMPs), in particular ADAMs (a disintegrin and metalloproteinases) such as ADAM10 and ADAM17 [12].

ADAM17 was found to be overexpressed in response to inflammation. In neuroinflammation, it has been reported to be an activator, via proteolysis, of the membrane precursor of TNFα, a cytokine that has been clearly correlated with AD [13]. For example, the receptors for IL-1 and IL-6 are enzymatically processed by ADAM17. ADAM17, in addition to having effects on the processing of cytokines and chemokines, is implicated in the cleavage of growth factors. For example, TGFα is generated by processing by ADAM17 as a product of TNFα [13].

Both ADAM17 and ADAM10 act on α-secretase, cleaving the amyloid precursor protein APP and generating soluble and non-amyloidigenic fragments, exerting a neuroprotective effect by inhibiting the formation of amyloid beta fragments (Aβ) [14]. Other enzymes involved in AD are also β-secretase and γ-secretase; the last one mediates the final cleavage and release of Aβ, on which clinical studies have been carried out [15]. However, the use of γ-secretase inhibitors has revealed the inhibition of the Notch signaling pathway, so the modulation of γ-secretase over inhibition is clinically preferred, with less production of Aβ fragments [16].

In this work, we went to evaluate how the cerebrospinal fluid, rich in the sCX3CL1 form, characterized based on canonical markers of Alzheimer’s disease in a previous work [17], from Alzheimer’s and non-Alzheimer’s subjects (other neurodegenerative pathologies) influenced the expression of specific markers on an “in vitro” co-culture of neurons–astrocytes–microglia (NAM).

## 2. Results

### 2.1. Effects of CSF from AD and Non-AD Patients on NAM Co-Culture

CSF samples from AD patients and non-AD patients were used as evaluation elements on morphological effects in a neurons–astrocytes–microglia (NAM) co-culture system.

The CSF samples used were selected from those analyzed in a previous study we conducted [17]. As reported in Table 1, the three different groups, Aβ+Tau, Aβ-Tau, and non-AD (four patients per group), were selected to have a very low variability compared to the average value of the different markers used for their characterization biochemistry of being AD or non-AD subjects.

The NAM co-cultures were treated with CSF from eight AD patients (four patients belonging to the group Aβ+Tau and four patients to the group Aβ-Tau) and four non-AD patients (group non-AD) while the control wells were treated with the same volume of CSF used by complete sterile PBS. After 24 h and 48 h of incubation time with CSF or PBS, the morphology of the co-cultures was observed through optical microscopy.

As shown in Figure 1 and quantified in Table 2, at time 0, the astrocytes had a starry morphology with different extensions and had formed a support network for the overlying neurons; the neuronal cells had aggregated, forming multiple neurospheres. After 24 h of incubation, no consistent differences were observed in the co-cultures treated with CSF from AD patients (groups Aβ+Tau and Aβ-Tau) compared to that from non-AD patients (group non-AD) and co-cultures not treated with CSF, used as a control (Control). Meanwhile, at 48 h of incubation, the same morphology appeared for the co-cultures treated with the CSF of patients of groups Aβ-Tau, non-AD, and Control in the co-cultures, while the neural network of the co-cultures treated with the CSF of patients of group Aβ+Tau had missing, substantially and completely, neurofilaments and spheroid, and many cells were in suspension. This suggested the presence, in the CSF of AD patients of group Aβ+Tau, of elements having destabilizing effects on neurofilaments, cellular adhesion processes, and intercellular contacts.

### 2.2. Gelatinolytic Activities and Their Quantification in CM and Cell Extracts from NAM Co-Cultures Treated or Not with CSF

An in vitro evaluation, in NAM co-culture models, of the proteolytic cascade was carried out, following the CSF treatment of AD subjects and non-AD subjects, through gelatin zymography experiments (Zy) to identify proteolytic activities, especially of enzymes belonging to the MMP family. The zymography results are shown in Figure 2 (a representative image of the different samples; in Appendix A, the zymographies of all the samples analyzed are shown); although preliminary, these indicate the activation of the cascade of MMPs by elements preferentially present in the CSF of AD and non-AD subjects in the CM and protein extracts.

The results also showed a difference in MMP expression based on CSF incubation time (24 and 48 h). As a control, a zymographic analysis with CSF alone of AD and non-AD patients was performed to evaluate the baseline expression of MMPs present in the CSF (Figure 2 and Appendix A).

For a correct evaluation of the gelatinolytic activities observed through the zymographies carried out with the different samples, these were evaluated in the integrated-density grey color (Image J64 program). In this type of analysis, for a broader understanding/generalization, samples Aβ+Tau and Aβ-Tau were combined as AD subjects.

As shown in Figure 3, the collagenolytic activities identified in the CMs of the different subjects showed a greater expression in the first 24 h of treatment in AD patients compared to non-AD patients, an effect that was more or less canceled after 48 h. No analysis was performed on the zymograms of the CM developed in the presence of EDTA as there were no collagenolytic activities.

The gelatinolytic activities identified and quantified in relation to the control (PBS-treated cells) were statistically evaluated according to the Mann–Whitney U-test (Table 3), as described in Section 4.10, for which the null hypothesis could be rejected for several samples, but not all.

The same evaluation, of integrated density, was also conducted on the collagenolytic activities present in the extracts (see M&M) of NAM in the co-culture, treated with the CSF of AD subjects (Aβ+Tau and Aβ-Tau) and non-AD subjects (non-AD), and developed in the presence of CaCl_2_ or EDTA.

As shown in Figure 4a, the Ca^2+^-dependent collagenolytic activities identified per m.w., but not per species (characterization studies are in progress), were, in general, more represented in AD samples at both 24 and 48 h compared to non-AD samples. Similarly, the analysis of the collagenolytic bands obtained in the presence of EDTA (Ca^2+^-independent proteases), as shown in Figure 4b, also identified by m.w. but not by species, presented a behavior similar to that previously described, that is, a greater enzymatic activity in the AD samples at both 24 h and 48 h compared to the non-AD samples.

Also in this case, the gelatinolytic activities identified both in the zymographies developed in the presence of Ca^2+^ and EDTA were quantified compared to the mean of the medians in relation to the control (cells treated with PBS) and statistically evaluated according to the Mann–Whitney U-test (Table 4), as described in Section 4.10. For this evaluation, the null hypothesis could be rejected for several samples, but, even in this case, not all.

### 2.3. Immunofluorescence Staining Using Different Antigens on NAM Co-Cultures Treated with CSF from AD (Aβ+Tau and Aβ-Tau), Non-AD (Non-AD), and PBS (Control) Subjects

The NAM co-cultures, after treatment with the CSF of the different subjects or with the same volume of PBS (Control) for 24 h, were fixed in formaldehyde and analyzed via immunofluorescence. In particular, the following antibodies were used for the specific antigens: anti-CX3CL1, anti-CX3CR1, anti-p38, anti-β-catenin, anti PECAM, anti-glial fibrillary and anti-CD11b fibrillar acid protein.

As reported in Figure 5, a low expression of CX3CL1 was observed in AD subjects (Aβ+Tau and Aβ-Tau) compared to controls and non-AD subjects while a higher expression of P38 was observed in AD subjects, especially in those treated with CSF from patients Aβ-Tau, also, but with lower evidence in the Aβ+Tau group. The CX3CR1 receptor did not seem to have substantial exchange on all analyzed conditions such as PECAM, a surface-specific antigen. Meanwhile, β-catenin showed a consistent decrease in expression in NAM treated with the non-AD CSF.

### 2.4. Quantitative PCR Performed on NAM Co-Cultures Treated with CSF from AD (Aβ+Tau and Aβ-Tau) and Non-AD (Non-AD) Subjects at 24 h and 48 h, Respectively

qPCR analyses on mRNAs extracted from NAM co-cultures of AD and non-AD subjects were conducted, using the *Gapdh* coding gene for housekeeping and normalizing the values to restrict the gene expression in NAM co-cultures treated with PBS (negative control). As shown in Figure 6, we chose genes recognized to exert a functional role in brain inflammation including pro- and anti-inflammatory genes (*Il1a*, *Il1b*, *Il6*, *Il10*, *Nfkb*, *Tnf*, *Tgfb*, *P38*, *Cx3cl1*). Similarly, genes encoding members involved in tau pathophysiology (*β-catenin*, *Cdk5*, *Gsk3b*, *Prkca*, *Gnaz*, *Gnao*) and amyloidogenesis (*Lrp1*, *App*, *Psen1*, *Psen2*, *Bace1*, *Bace2*, *ApoE*, *Fmrp*, *Abca*, *Apbb1*, *Apbb2*, *A2m*, *Aplp2*, *Aph1a*, *Hnrpc*) were transcriptionally analyzed. In addition, transcriptional changes in genes encoding MMPs (*Mmp2*, *Mmp3*, *Mmp7*, *Mmp8*, *Mmp9*, *Mmp10*, *Mmp11*) were also profiled.

From the heatmap in Figure 6, it can generally be seen that in both AD and non-AD subjects, there was an upregulation of almost all the genes involved in the inflammatory process, especially after 48 h of treatment, except for *P38* and *Akt*, where it appears that treatments resulted in negligible effects on *P38* transcriptional levels while *Akt* was downregulated.

Different gene regulation profiles emerged when comparing genes regulating the hyperphosphorylation of tau in NAM co-cultures treated with AD and non-AD CSF. In particular, upregulation was observed in all the analyzed genes from AD-CSF-treated samples, with the exception of *Gsk3β* and *βcat,* which were downregulated at 24 h and 48 h, respectively. Interestingly, the *Gsk3β* mRNA levels increased at 48 h and the *β-cat* coding gene was mostly downregulated, both in AD and non-AD subjects. Conversely, treatment with CSF from non-AD subjects provided different expression profiles because of a general downregulation at 24 h and slight *βcat*, *Prkca,* and *Gnao* mRNA level increases at 48 h while the others were unaffected or remained downregulated.

Among the gene set that was amyloidogenic-related, we found that for *App* and *ApoE,* there was no marked change compared to the control condition (NAM co-cultures treated with PBS) either at 24 or 48 h. Differently, in the co-culture treated with CSF from AD subjects, at 48 h, there was an overexpression of all the other genes implicated in amyloidogenesis. In NAM co-cultures treated with CSF from non-AD subjects, only *Bace1*, *Bace2*, *Apbb2*, and *Hnrpc* were upregulated at 48 h; all the others, however, were downregulated when compared to the control.

Also, members of the MMPs gene family were differently affected. Transcriptional profiling of NAM co-cultures treated with CSF from AD subjects showed that, exclusively, *Mmp3*, *Mmp8,* and *Mmp11* resulted upregulated at 24 h whereas as a general rule, all the *Mmps* were upregulated at 48 h. A more pronounced downregulation was observed in co-cultures after treatment with non-AD CSF with the exception of *Mmp2*, *Mmp8,* and *Mmp10* levels (unchanged) and *Mmp11* (upregulated) at 24 h. As had previously occurred, with the exception of *Mmp9,* all the members were upregulated at 48 h.

## 3. Discussion

From what has been reported regarding the effects induced by the cerebrospinal fluid of AD and non-AD subjects, we observed, from a morphological point of view, the loss of morphology in NAM co-cultures. In particular, this was the case when referring to the neurospheres when treated with CSF from AD subjects of group Aβ+tau; in the Control group, group Aβ-tau, and group non-AD, any morphological modification was not observed. This suggested that in the CSF of Aβ+tau subjects, there are elements implicated in the structural organization of neurons, elements having destabilizing effects on neurofilaments, cellular adhesion processes, and intercellular contacts.

We, therefore, analyzed, both in the CM and in the extracts from NAM co-cultures treated with CSF from AD subjects (groups Aβ+tau and Aβ-tau, together) and non-AD patients (group non-AD), the presence of gelatinolytic activities via zymography developed in the presence of CaCl_2_ (activator of MMPs and other Ca^2+^-dependent proteolytic enzymes) or EDTA. The proteolytic activity amount was determined through the integrated-density grey color (Image J64 program (https://download.cnet.com/imagej-64-bit/3000-2192_4-75879092.html (accessed on 1 June 2010))). Greater and high numbers of collagenolityc activities were present in the CM and in the cell extracts from the NAM co-cultures (CaCl_2_) in AD subjects compared to non-AD subjects. Furthermore, in the CM of AD subjects, there were two bands of 58 kDa and 57.5 kDa, of which the first could be traced back to MMP8 while the second could be traced back to MMP1, MT1MMP (soluble form), or MMP10. Moreover, in the CM of AD subjects, there were two bands of 58 kDa and 57.5 kDa, of which the first could be traced back to MMP8 while the second could be traced back to MMP1, MT1MMM (soluble form), or MMP10.

Also, in the cell extracts obtained from the AD samples, there were four proteolysis bands of 144.5 kDa, 130 kDa, 75 kDa, and 74 kDa, which, at the moment, we do not know to what we can compare.

Also, zymographies, developed in the presence of EDTA, of extracts from NAM co-cultured cells treated with CSF from AD subjects showed gelatinolytic, Ca^2+^-independent activities in higher numbers than those detected in extracts obtained after treatment with the CSF of non-AD subjects. In particular, two bands of 79 kDAa and 76 kDa were present, but even in this case, we are currently unable to trace them back to specific enzymes.

In immunofluorescence assays on NAM co-cultures treated with CSF from AD subjects, we observed a decrease in the expression of CX3CL1, and with less evidence, in CSF from non-AD subjects compared to the control. This was an effect that could be traced back to sCX3CL1 generation by proteolytic enzymes present in the CSF. On the other hand, an increase in P38 was observable in slides treated with CSF from AD subjects compared to the control (treated with PBS only) and the subjects treated with CSF from non-AD patients.

The reported data are supported regarding the expression of the messengers extracted from the co-cultures treated with CSF from AD and non-AD patients and compared to their basal expression in the co-cultures (mRNA extracted after treatment with PBS, same volume of CSF).

As reported in Figure 1, the expression of the different mRNAs analyzed would be very different if the co-cultures were treated with CSF from AD or non-AD subjects.

It is known that inflammation is a central mechanism in Alzheimer’s disease and the activity of anti-inflammatory cytokines is crucial in maintaining a delicate balance of immune responses to prevent the onset of chronic inflammatory diseases [18,19,20]. Therefore, it is not unexpected that there will be an upregulation of *Il10* and *Tgfb,* which likely represents a tentative means to cope with the aberrant overexpression of the inflammatory pathway members herein studied.

Among different mechanisms regulating the pathophysiology of the brain diseases, Wnt/β-catenin signaling is an essential signal transduction pathway that has emerged as a central pivot for the regulation of multiple different pathways in Alzheimer’s disease (AD) pathogenesis including neuronal survival, the inhibition of *Bace1* expression, and the suppression of Aβ production [9,21,22]. Moreover, it is widely accepted that the triggering of *Wnt/β-catenin* signaling leads to a reduction in *Gsk3β* activity, resulting in the subsequent inhibition of tau phosphorylation [23,24]. Pathway-focused gene expression analyses showed concordant *Gsk3β* upregulation and *β-catenin* downregulation, especially at 48 h, thus suggesting the onset of mechanisms leading the suppression of *Wnt/β-catenin* signaling in NAM co-cultures treated with AD CSF. The qRT-PCR analyses also showed the upregulation of *Cdk5* and *Prkca,* which are involved in the pathogenesis of Alzheimer’s disease [25,26] via tau phosphorylation in several sites. All this evidence is likely to support the hypothesis for the accumulation of intracellular pathogenic NFT tangles. However, since a different gene expression pattern was achieved in NAM co-cultures treated with non-AD CSF, it appears that such possibilities did not occur in the absence of AD CSF.

Consistently, also, the overexpression of genes belonging to the amyloidogenic pathway, recently discussed by Hampel et al. [27], as herein studied represents a proxy for the promotion of a pathogenic background leading to neuronal degeneration.

In recent years, several lines of evidence have supported the contribution of MMPs within the pathophysiological mechanisms of AD, especially due to interactions with Aβ or tau [28,29,30,31,32].

*Gsk3* is upregulated in AD while it is downregulated in non-AD, a finding supported by the expression of *β-catenin* (negative regulator of *Gsk3*). It is also supported by the expression of *Nfkb*, an activator of ROS neurons, characteristic of AD patients, although it is also upregulated in animal cells treated with CSF from non-AD subjects, where an inflammatory effect is nevertheless observed.

It is well known that in the pathology of Alzheimer’s disease, there is a condition of generalized and progressive inflammation during its evolution, both occurring in the brain system, then endothelial cells expressing CX3CL1 and its receptor CX3CR1. On the other hand, the condition of hypoxia linked to tissue inflammation induces the release by the proteolytic action of sCX3CL1. A clustering in specific membrane domains of a whole series of proteolytic enzymes, in particular belonging to the family of extracellular matrix metalloproteases (MMPs), has been highlighted, with the concerted action of these enzymes in the digestion and arrangement of the ECM and the activation of a proteolytic cascade [33].

According to the results, treatment with CSF from AD and non-AD subjects induces the overexpression of several MMPs. The unidentified components in the CM of NAM cultures could be traced back to the 58 kDa of MMP8 and the 57.5 kDa form MMP1 or MT1-MMP (soluble form). Effectively, the qRT PCR showed the upregulation of *Mmp8* and *Mt1mmp* (*Mmp11*) mRNA; however, we do not have information on *Mmp1* and the downregulation of Mmp10.

There is no doubt that in the analyses conducted on the effects of CSF from AD and non-AD subjects, an important inflammatory response was activated by the co-culture. The different behavior under the action of CSF from AD and non-AD subjects raises several questions: (i) are different forms of sCX3CL1 activating different responses? (ii) Which cellular components are targeted by sCX3CL1, and are they diversified? Many other questions can be raised. But the fundamental question is: is sCX3CL1 really the “conductor” underlying the observed inflammatory response?

## 4. Materials and Methods

### 4.1. CSF Sample

Patients with suspected clinical cognitive decline (CD) were evaluated by the Center for Dementia and Cognitive Disorders at the Neurology Unit, University Hospital of Palermo. For each patient, history was taken and a general physical and neurological examination was performed. Patients then underwent a cognitive/behavioral assessment to clinically confirm and define the suspected CD. Caregivers underwent a structured interview to quantify the severity of dementia: CDR (Clinical Dementia Rating) [34]. Once CD was confirmed, patients underwent structural brain imaging (magnetic resonance imaging) to rule out any secondary causes of CD (such as vascular disease or malignancy) and to evaluate regional brain atrophy and brain imaging (PET) [35] to evaluate regional brain glucose metabolism. A lumbar puncture (LP) was performed to collect cerebrospinal fluid (CSF) [36]. The CSF was collected as a routine procedure in two different tubes: one for standard physical and chemical analysis and a second, in polypropylene, for neurochemical analysis. All CSF samples were centrifuged, aliquoted, and stored at −80 °C. A chemical–physical examination of the cerebrospinal fluid (cell count, protein, and glucose concentration) was performed to exclude an inflammatory/infectious disease in the brain that could cause CD. Finally, specific levels of CSF biomarkers for AD (Aβ1–40, Aβ1–42, Ratio Aβ42/40, t-tau, and p-tau) were determined according to clinical biochemistry. These CSF samples obtained from AD and non-AD patients were analyzed for the presence of the soluble form of CX3CL1, together with the canonical markers Aβ1–40, Aβ1–42, and Ratio Aβ42/40, together with t-tau and p-tau, via ELISA and CLEIA test. Therefore, 3 groups were generated and defined as Aβ+Tau and Aβ-Tau, AD patients; and non-AD subjects with other different neuropathologies. Group Aβ+Tau consisted of 16 patients characterized by an expression of Aβ forms and ratio, together with both total and phosphorylated tau in the canonical representation for AD subjects; group Aβ-Tau included 12 patients characterized by amyloidosis, where the amyloid fibers were well represented; however, in the group Aβ-Tau, both total tau and phosphorylated Tau (p-Tau) were around normal values; however, they represented, together with Aβ+Tau group, by clinical evaluation, the AD subjects, i.e., sick. The third group, non-AD, included 18 subjects defined by us as non-AD, based on characteristics, since the analyzed markers were in a range of non-Alzheimer conditions, although clinically subject to different neuropathologies. The results obtained highlighted the presence of CX3CL1 for a quantity greater than 40% in subjects with AD compared to non-AD subjects. This suggests that CX3CL1 could be considered a possible new marker for Alzheimer’s in the disease [17].

### 4.2. Evaluations of Aβ 1–42, Aβ 1–40, Aβ 42/40 Ratio, Tau-Total, Tau-Phosphorylated, and CX3CL1 in Selected CSF Were Performed via Chemioluminescence Enzyme Immunoassay (CLEIA) and ELISA in the Selected Groups of Aβ+Tau, Aβ-Tau, and Non-AD Patients

The CSF of Aβ+Tau, Aβ-Tau, and non-AD subjects was analyzed for the presence of *Aβ 1–42, Aβ 1–40,* total tau (Tau-total), and hyperphosphorylated tau (Tau-phosphorylized) via CLEIA. The analyses were conducted using Lumipulse G Total Tau and Lumipulse Gp Tau 181 (in hyperphosphorylated tau identification), from Fujirebio Inc. Europe, Gent, Belgium according to the instructions specified by the vendor. Meanwhile, the presence of CX3CL1 in the different groups was obtained via a sandwich-enzyme-linked immunosorbent assay using the Wuhan Fibe Biotech Co. Kit (Wuhan, China) according to manufacturer’s instructions.

### 4.3. NAM Co-Cultures

Neurons were purified from the cerebral cortexes of rat fetuses at day 16 of gestation and co-cultured with astrocytes and microglia (5 × 10^4^/cm^2^) in a serum-free Maat medium [37,38,39].

The cerebral hemispheres of 16-day-old rat fetuses were removed via aseptic surgical procedure, placed in a falcon containing 20 mL of Hams F12 + 20% NCS (newborn calf serum), and cleaned of their meningeal covers under a dissecting microscope. The tissue was fragmented through a series of passes through a Pasteur pipette and then dissociated via filtration through a nylon sieve (50 µm) by gently applying pressure using a rubber “scraper”. To remove aggregates and cerebral vessels, the cell suspension was again filtered through another nylon sieve (25 µm). After centrifugation at 250× *g* for 5 min, the cell pellets were resuspended in F12, washed twice, and, finally, resuspended in 5–10 mL of Maat medium and plated at 2.5 × 10^5^/cm^2^.

Obtained NAM co-cultures were treated with CSF from 8 AD patients (4 patients belonging to group Aβ+Tau and 4 patients to the group Aβ-Tau) and 4 non-AD patients (non-AD) while the control wells were treated, with the same volume of CSF used, in completely sterile PBS. For applications (immunofluorescence, zymographic analysis, and qPCR), at least three wells were generated for each type of treatment (see below). The NAM co-culture was incubated with CSF samples or PBS at different times: 24 h and 48 h.

At least three different experiments were conducted for each type of evaluation.

### 4.4. Morphological Assay

Neuron cells were co-cultured with astrocytes and microglia (5 × 10^4^/cm^2^), at a density of 2.5 × 10^5^/cm^2^ cells per well, in a 24 multiwell. Three wells were generated for each sample, therefore: 12 treated with CSF from 4 Aβ+Tau subjects, 12 treated with CSF from 4 Aβ-Tau patients, 12 treated with CSF from 4 non-AD people, and 12 developed in the presence of complete PBS as a control. Phase contrast images (Carl Zeiss™ Axio Vert.A1 FL-LED Inverted Microscope, Milan, Italy) of each well were acquired, and the number of neurospheres per photographic acquisition was counted and the average of these determined. Furthermore, the average surface area occupied by the neurospheres was determined for each well, considering their heterogeneity in size.

Three different experiments were performed for each type of evaluation.

### 4.5. Immunofluorescence (IF) Assays

Neuron cells were co-cultured with astrocytes and microglia (5 × 10^4^/cm^2^), at a density of 2.5 × 10^5^/cm^2^ cells per well, in a 24-multiwell slide containing sterile coverslip in Maat medium. Three coverslips were generated for each sample, therefore: 12 treated with CSF from 4 Aβ+Tau subjects, 12 treated with CSF from 4 Aβ-Tau patients, 12 treated with CSF from 4 non-AD people, and 12 developed in the presence of complete PBS as a control. After 24 h of growth at 37 °C, cell lines on the slides were fixed with 3.6% formaldehyde for 10 min and rinsed with PBS three times. The cells were permeabilized via treatment with a 0.2% Tron X-100 solution in PBS for 15′. Then, non-specific reaction sites were blocked by adding a 3% BSA solution in PBS for 2 h at r.t. At this point, the cells were labeled with the following primary antibodies diluted in PBS overnight at 37 °C: CX3CL1 antibody (1:80, Abcam, Cambridge, UK), CX3CR1 antibody (1:50, Abcam), p38 antibody (1:100, Abcam), β-catenin antibody (1:500, SIGMA Mialn, Italy), PECAM antibody (1:100, SIGMA), anti-glial fibrillary (1:80, SIGMA) antibody, and anti-CD11b antibody (1:80, SIGMA). Following 3 washes in PBS, the cells were labeled with the following secondary antibodies diluted in PBS for 2 h at 37 °C: Mouse Texas Red (1:250, SIGMA), Mouse FITC (1:250, SIGMA), Rabbit Texas Red (1:250, SIGMA), and Rabbit FITCH (1:250, SIGMA), in specific association with primary antibodies used. In some of the cells, the actin cytoskeleton was labeled with phalloidin FITC (1:100 in PBS, SIGMA) for 5 min at room temperature. After 3 washes with PBS, cells were stained with DAPI (1:20.000 in PBS, SIGMA) for 10 min, at room temperature, for marking the nuclei in blue. Finally, the NAM slides were mounted and observed through epifluorescence microscope (Leica, Wetzlar, Germany). Experiments were performed three times.

### 4.6. Extraction of Soluble Proteins from NAM Co-Culture Cells

Neuron cells were co-cultured with astrocytes and microglia as previously described in immunofluorescence experiments (without coverslips). After treatment with the different CSFs or PBS, the conditioned medium was taken (conditionated medium, CM) and the co-cultures were washed with PBS and without Ca^2+^ and Mg^2+^; detached by Trypsin-EDTA 1× in PBS, cells detached by 3 different wells of each type of treatment were collected and impelled using centrifugation at 1000 rpm for 5′. The pellets were resuspended in an appropriate volume (depending of the pellet size) of RIPA Buffer 1X (50 mM Tris-HCl pH 7.5; 0.5% sodium deoxycholate; 150 mM NaCl; 1% Triton X-100) supplemented with protease inhibitors (1 mM of phenylmethylsulfonyl fluoride (PMSF); 1 µM of Pepstatin A; 100 µM Leupeptin; 10 mM Ethylenediaminetetraacetic acid (EDTA)). Therefore, samples were incubated in ice for 15 min and constantly subjected to micro vortex mixer treatments, then centrifuged at 10,000× *g* for 20′. The number of proteins extracted and presented into the supernatant was calculated via the Bradford assay (Bradford Reagent, Sigma-Aldrich) using different concentrations of bovine serum albumin (BSA) as standard. A quantity of 10 μL of each conditioned medium of NAM co-cultures and 8 ng/sample of proteins extracted were mixed with sample buffer 1X (62.5 mM Tris-HCl pH 6.8; 2.5% SDS; 0.002% Bromophenol Blue; 10% glycerol) and immediately situated in ice and analyzed via zymography assay. Experiments were performed three times.

### 4.7. Zymography Assay

After treatment with the different CSFs or PBS, the conditioned medium was taken (conditionated medium, CM). The CMs from 3 wells of each treatment were collected to standardize the sample. The co-culture cells were washed with PBS without Ca^2+^ and Mg^2+^; detached by Trypsin-EDTA 1× in PBS, cells detached by 3 different wells of each type of treatment were collected and impelled using centrifugation at 1000 rpm for 5′.

The proteins were separated by 7.5% sodium dodecyl sulfate Polyacrylamide gel electrophoresis (SDS-PAGE) that was gelatine-supplemented. After electrophoresis, gelatine zymographies were extensively washed in Tris-HCl (50 mmol/L; pH 7.4) containing TRITON X-100 2% to eliminate the excess of SDS. Then, they were incubated overnight at 37 ◦C in two different developing buffers: activator buffer containing 2 mmol/L CaCl_2_, Tris-HCl buffer (50 mmol/L; pH 7.4), 1.5% Triton X-100, and 0.02% Na_3_ Azide to activate gelatinases; or inhibitor buffer containing 2 mmol/L EDTA and Tris-HCl buffer (50 mmol/L; pH 7.4) containing 1.5% Triton X-100 and 0.02% Na_3_ Azide to inhibit gelatinase activities. After incubation, gels were stained using Coomassie Brilliant Blue R-250 (Sigma).

### 4.8. RNA Isolation and cDNA Synthesis

Total RNA was extracted from NAM co-cultures using Trizol reagent (Invitrogen, Carlsbad, CA, USA). Also, in this preparation, 3 wells per preparation were collected together. RNA concentrations and quality were verified using spectrophotometry (optical density (OD) at 260 nm) whereas the RNA integrity was checked using a 1.5% agarose gel. The RNA was stored at −80 °C for future use. The extracted RNA (500 ng) was treated with RNA qualified 1 (RQ1) RNase-Free DNase (Promega, Madison, WI, USA) to remove any residual genomic DNA contamination, and the DNase was inactivated by adding 25 mM EDTA. First-strand cDNA was synthesized from 250 ng DNase-treated total RNA samples using random primers and High-Capacity cDNA Reverse Transcription Kit (Life Technologies Corporation, Carlsbad, CA, USA), following the manufacturer’s instructions. The cDNA mixture was stored at −20 °C.

### 4.9. RT-qPCR

RT-qPCR was performed using the BIO-RAD CFX96 System with Power Sybr Green as the chemical detection method (Applied Biosystems, Forster City, CA, USA). Real-time PCRs were carried out in 96-well plates in a 20 µL mixture containing 1 µL of a 1:20 dilution of the cDNA preparations, using the following PCR parameters: 95 °C for 10 min, followed by 40 cycles of 95 °C for 10 s and then 60 °C for 60 s. The sequences of the specific primer pairs used for qPCR are shown in Table 5. Samples were run in duplicate. The GAPDH was chosen as reference gene and used to quantify the expression levels of the target genes. Experiments were performed three times.

### 4.10. Densitometry Evaluation of Protein Expression

To quantify different enzyme expressions selected by zymography gels, the images were evaluated in integrated-density grey color by Image J64 program. Evaluation was performed on SDS-PAGE or via zymography separation for each sample.

### 4.11. Statistical Analyses

The data obtained were evaluated for normality by applying the Shapiro–Wilk test. The variance that occurred between the CSFs of different groups analyzed was determined according to one-way ANOVA (*p*-value).

For the other analyses, see the limited number of samples used in the different experiments, for which a canonic Student’s t-test could not be used below. The Mann–Whitney U-test (nonparametric test) was applied. In particular, the medians between the Aβ+Tau, Aβ-Tau, and non-AD objects were evaluated compared to the control (co-cultures of NAM treated with PBS). Additionally, the values of the different bands in zymography, obtained from the evaluation via integrated intensity (Image J64) of the different samples, were compared to the band values of the standards. The null hypothesis (HO—equality of values between the two groups) was rejected for *p* < 0.05.

## Data Availability

Data is contained within the article and Appendix A.

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
