# Peer review of "Effects of Cerebrospinal Fluids from Alzheimer and Non-Alzheimer Patients on Neurons–Astrocytes–Microglia Co-Culture"

_ijms, 2024, doi:10.3390/ijms25052510_

Round 1

Reviewer 1 Report

Comments and Suggestions for Authors

In the articles, Ghersi, Iemmolo and colleagues mostly focus on using cerebrospinal fluid, taken from patients with Alzheimer’s disease (with or without Tau pathology) or control individuals, and testing its effect on a neuronal co-culture system. The general approach is very interesting, but there are significant technical issues which must be addressed.

The most significant of these is a complete absence of statistical analysis. The authors draw various conclusions from their work, but it appears that no attempt has been made to determine if effects are statistically significant or not.  It is also unclear in most cases how many times experiments have been replicated.  Only significant results should then be shown in scheme 1. Without complete clarity on replication strategy and whether results are consistent/significant, this work cannot be published.

It is also highly unusual that the authors end with another set of work (a rat animal model for AD) that is seemingly unconnected to the main elements of the paper.  They even give it a second title. Unless the authors include experiments linking these two elements (e.g. testing the CSF on this model), then I think it is much better to delete the model from this manuscript and publish it separately.

The group names (A’, A’’ and B) are not intuitive.  A naming scheme that clarifies what they are (e.g. AB+Tau, AB-Tau, Control) would make the paper much easier to follow.

Data in figure 1 seems very subjective and open to interpretation.  A quantitative study, conducted by a researcher blinded to the diagnostic status of each sample, should be performed to confirm the result without possibility of experimental bias.

Clearer (quantitative) definitions are required for the levels of Tau (and phosphorylated Tau) used to define groups A’ and A’’ – maybe in the form of a table

Size markers are not shown on Western blots (figure 7B), and full images of the blots should be supplied for review purposes

Minor issues

In some figures (e.g. figure 4), samples are labelled as “AD”, and it is unclear if this refers to group A’, A’’ or a combination or both.

Company names and order codes should be provided for all antibodies used

It is unclear what “almost 3-gel SDS-PAGE” means (line 505)

The text in figure 3 is very small and should be enlarged for clarity.

In several instance (e.g. line 114) incorrect symbols are used – probably a formatting error

Comments on the Quality of English Language

The English is fully comprehensible, but with numerous minor errors. A few examples from the abstract;

Line 23: Add a hyphen for “non-AD”

Lines 24-26: The two sentences should be merged into one

Line 27: “fro” instead of “for”

Line 28: “values should be changed to “quantified”

Author Response

Note are in attached file

Reviewer 2 Report

Comments and Suggestions for Authors

The authors describe an interesting study on the influence of cerebrospinal fluid from Alzheimer's patients on neuronal cell cultures. Another task of the manuscript is to generate an animal model of Alzheimer's disease. Both studies look like independent experiments. Even in the title, the authors separate these tasks. I think some more experiments could connect these two parts of the study. For example, the expression of CX3CL1 forms and inflammatory markers in the brain of model rats could be added to the article. 

In the introduction, the authors pay most attention to the role of CX3CL1 in AD pathogenesis. However, the main research topic of the article is the expression of different matrix metalloproteases (MMPs). The article would benefit from a more detailed discussion of the role of MMPs in AD pathogenesis.

There are major comments:

1. The authors state that the neuronal co-cultures treated with AD CSF lose neurofilaments and spheroid structure. Figure 1 is not representative to assess these changes. Immunofluorescence staining for neurofilaments and other markers (such as synapsin) would help to illustrate the changes.

2. Figure 5 is of poor quality and the immunostaining appears non-specific. This may be due to methodological imprecision. According to section 4.3. Immunofluorescence (IF) assays, the authors forget to block non-specific antibody binding with BSA solution. In addition, permeabilization is necessary for immunostaining intracellular proteins (most of the antigens studied, except CD11b and possibly CX3CR1).

3. Is it possible that the q-PCR analysis of gene expression showed different results due to the different proportions of neurons, astrocytes and microglia in the co-cultures? The primers for Gapdh are not listed in materials and methods.

4. The legend for Scheme 1 is not informative. The quality of the figure is poor.

Minor points:

1. Acronyms must be deciphered at the first mention in the text (MMPs, CM).

2. Rat gene names should be written in italics with the first letter capitalised.

3. Genes with increased expression levels are usually referred to as 'upregulated', but not 'overexpressed'. The latter term is used for transgenes.

4. References 17, 33-35 are incomplete. The journal name is missing.

Comments on the Quality of English Language

Moderate English editing is required. Some long sentences with multiple semicolons are difficult to read.

Author Response

Note are in the attachment

Round 2

Reviewer 1 Report

Comments and Suggestions for Authors

The authors have replied to the vast majority of my concerns very well, and I commend them on the thorough addition of statistical analyses in particular.  My one outstanding concern is regarding figure 1 which, as agreed by the authors, is of poor quality.

To remove the subjectivity of this, I ask the authors to do the following:  Collect all of the images (data) taken of these cells, and anonymize (blind) them, so that it is not possible to know the diagnostic status and age. Then ask a researcher, who is unfamiliar with the data to give assessments of the morphology using a numerical scale.  Then unblind the data and determine whether there are significant differences or not between the different diagnostic statuses.

Author Response

The authors have replied to the vast majority of my concerns very well, and I commend them on the thorough addition of statistical analyses in particular.  My one outstanding concern is regarding figure 1 which, as agreed by the authors, is of poor quality.

To remove the subjectivity of this, I ask the authors to do the following:  Collect all of the images (data) taken of these cells, and anonymize (blind) them, so that it is not possible to know the diagnostic status and age. Then ask a researcher, who is unfamiliar with the data to give assessments of the morphology using a numerical scale.  Then unblind the data and determine whether there are significant differences or not between the different diagnostic statuses.

As suggested, and we thank the reviewer for this, we showed the "blind" images to a colleague totally unrelated to this experiment (even to the research field), his attention was actually focused on the number and different sizes of the neurospheres in the different samples. From these observations we generated Table no. 2 which relates the number of neurospheres and the surface they occupy in the different samples to integrate the image. And commented on the difference.

Reviewer 2 Report

Comments and Suggestions for Authors

The manuscript has been greatly improved. Some minor points:

1. Lines 87-104. The newly added text lacks references. Should be added.

2. Lines 199-200 and 229-230: It is better to briefly describe the cases where parameters differ significantly between samples.

Author Response

The manuscript has been greatly improved. Some minor points:

  1. Lines 87-104. The newly added text lacks references. Should be added.

It has been done

2. Lines 199-200 and 229-230: It is better to briefly describe the cases where parameters differ significantly between samples.

It has been done